# Posterior Decompression and Fixation for Thoracic Spine Ossification: A 10-Year Follow-Up Study

**DOI:** 10.3390/jcm12175701

**Published:** 2023-09-01

**Authors:** Juntaro Maruyama, Takeo Furuya, Satoshi Maki, Takaki Inoue, Atsushi Yunde, Masataka Miura, Yuki Shiratani, Yuki Nagashima, Yasuhiro Shiga, Kazuhide Inage, Yawara Eguchi, Sumihisa Orita, Hiroshi Takahashi, Masao Koda, Masashi Yamazaki, Seiji Ohtori

**Affiliations:** 1Department of Orthopedics Surgery, Graduate School of Medicine, Chiba University, Chiba 260-8677, Japan; maruyama.juntaro@gmail.com (J.M.);; 2Center for Frontier Medical Engineering, Chiba University, Chiba 263-8522, Japan; 3Department of Orthopedic Surgery, Graduate School of Comprehensive Human Sciences, University of Tsukuba, Tsukuba 305-8577, Japan

**Keywords:** ossification of the posterior longitudinal ligament, posterior decompression with instrumented fusion, thoracic spine, long-term outcomes, kyphotic angle, additional surgery, patient follow-up

## Abstract

Ossification of the posterior longitudinal ligament of the thoracic spine (T-OPLL) causes symptoms including leg and back pain, and motor and sensory deficits. This study retrospectively reviewed 32 patients who initially underwent posterior decompression with instrumented fusion (PDF) for T-OPLL between 2001 and 2012, with 20 qualifying for the final analysis after applying exclusion criteria. Exclusions included unknown preoperative neurological findings, follow-up less than 10 years, or prior spinal surgeries at other levels. Outcomes were assessed using the Japanese Orthopedic Association (JOA) score, recovery rate, and kyphotic angle. The average preoperative JOA score of 3.6 improved to 7.4 at 1 year post-surgery and remained at 7.4 at 10 years, with a recovery rate of 52%. The kyphotic angle at T4–12 increased from 26 degrees preoperatively to 29 degrees postoperatively and to 37 degrees at 10 years. At the fused levels, the angle remained at 26 degrees immediately post-operation and increased to 32 degrees at 10 years. Forty percent of patients required additional surgery, primarily for conditions related to cervical OPLL, such as myelopathy, or lumbar OPLL, such as radiculopathy, or cauda equina syndrome. In conclusion, PDF effectively reduces T-OPLL symptoms over the long term, but the high rate of additional surgeries calls for careful patient follow-up.

## 1. Introduction

Ossification of the posterior longitudinal ligament of the thoracic spine (T-OPLL) is a relatively rare radiological finding. It is characterized by the replacement of ligamentous tissue by ectopic new bone formation, a pathological alteration that has been the subject of extensive research [1,2,3,4,5,6,7,8]. This change within the spinal structure can lead to the compression of the spinal cord or spinal nerves, resulting in a wide array of symptoms. Patients may experience pain, numbness, difficulty walking, and other neurological manifestations that can severely impair their quality of life. The complexity of T-OPLL requires careful consideration of treatment options, and surgical intervention is often needed when symptoms become severe or disabling. Among the various surgical approaches developed to address T-OPLL, posterior decompression with instrumented fusion (PDF) has emerged as an increasingly popular method. This preference is due to its positive results in reducing symptoms and its relatively low complication rate, as supported by numerous studies [1,3,9,10,11,12,13,14,15,16].

Despite the growing prevalence of PDF procedures in the treatment of T-OPLL, there remains a significant gap in our understanding of its long-term effectiveness. Much of the existing research has been concentrated on short-term outcomes, with only a handful of studies exploring the long-term effects of this procedure [3,17]. This lack of thorough investigation is particularly concerning, given that T-OPLL is often associated with systemic ossification tendencies. These tendencies can lead to the development of ossification in other levels of the spine, further complicating the patient’s condition [3]. The need for a more profound understanding of the long-term implications of PDF in T-OPLL patients is evident.

The primary objective of this study is to assess the long-term outcomes of PDF for patients with T-OPLL. By carefully examining the outcomes of T-OPLL patients who underwent PDF and have been followed up for an extended period of more than 10 years, this research aims to contribute valuable insights into the disease’s characteristics and the lasting effects of the PDF procedure and highlight specific factors to be observed in the long-term postoperative course of T-OPLL. The findings from this study will enable clinicians to closely monitor the patients’ postoperative course and adapt additional treatment as necessary, ultimately contributing to more effective care for those afflicted with this complex pathology.

## 2. Materials and Methods

### 2.1. Study Design and Patient Selection

The present study adopted a retrospective case series design, focusing on the long-term outcomes of PDF for thoracic ossification of T-OPLL.

The initial pool comprised 32 patients (17 men and 15 women) who underwent PDF for thoracic T-OPLL at our institution between September 2001 and December 2012.

The exclusion criteria for this study were carefully selected to provide the most relevant insights into the long-term outcomes of PDF for T-OPLL. Patients with unknown preoperative neurological findings (n = 5) were excluded to ensure that all included subjects had well-documented neurological findings before surgery, allowing for a more accurate assessment of surgical outcomes. Those with a final follow-up date less than 10 years after the first surgery (n = 6) were also excluded, as we aimed to include only those with more than 10 years of follow-up to enable a detailed examination of the long-term effects of the surgery. Additionally, those who had undergone other levels of spinal surgery prior to the first surgery (n = 1) were excluded to focus solely on the specific outcomes of PDF for T-OPLL. These criteria were instrumental in shaping the study’s focus and ensuring that the results were as applicable and insightful as possible. After applying these carefully considered exclusion criteria, a total of 20 patients were eligible for this study.

We collected extensive data on patient demographics, including age, sex, body mass index (BMI), and any pertinent comorbidities such as diabetes or hypertension. In addition to these demographics, we also gathered data on the preoperative JOA scores to assess the neurological status of the patients before surgery. Furthermore, we retrospectively reviewed the available medical records to determine the duration from symptom onset to the time of surgery for each patient. A flowchart, presented in Figure 1, illustrates the inclusion and exclusion criteria for the patients.

In this case series, the surgical approaches performed for T-OPLL were standardized to minimize variations in surgical technique. The selected approach involved fusion surgery two or three levels above and two or three levels below the stenotic lesion with OPLL, without performing kyphosis correction. To prevent neurological complications during decompression, a temporary rod was inserted. This measure was taken to provide structural support during the procedure, reducing the risk of damage to the spinal cord. Additionally, screw insertion was avoided at the most stenotic level to minimize the risk of neurological exacerbation. Furthermore, patients were thoroughly informed before the first surgery about the possibility of additional anterior decompression if postoperative improvement was found to be poor, as evidenced by a lack of change in neurological function (e.g., JOA scores remaining unchanged).

### 2.2. Clinical Outcomes

Clinical measures included the Japanese Orthopedic Association (JOA) score (excluding upper extremity scores) with a possible total of 11 points and an assessment of recovery rate using Hirabayashi’s method (recovery rate = [postoperative JOA score − preoperative JOA score]/[11 − preoperative JOA score] × 100) [18].

During the follow-up period, we also evaluated cases in which additional spinal surgery was required. The reasons for these additional surgeries included the presence of ossification of the ligaments at other levels, adjacent intervertebral disorders at the upper or lower instrumented vertebrae, and cases where postoperative improvement was considered insufficient, as indicated by a lack of change in neurological function (e.g., JOA scores remaining unchanged), requiring additional anterior decompression.

### 2.3. Radiographic Evaluation

The radiographic evaluation in our study included a detailed analysis of the kyphotic angles and morphology of ossification foci, crucial for understanding the long-term outcomes of PDF for T-OPLL. Specifically, the kyphotic angles in T4–12 and the fused level were measured using the Cobb method on lateral radiographs at three distinct time points: pre-operation, post-operation, and 10 years after surgery [19]. This time-series approach allowed us to track the progression of kyphotic angles, and we further investigated the correlation between the JOA score recovery rate and the kyphotic angle. Additionally, we conducted a thorough assessment of the morphology of the ossification foci. Utilizing CT scans, we classified the foci into six categories: linear, beaked, continuous waveform, continuous cylindrical, mixed, or circumscribed types, following the classification established by the Research Group for Ossification of the Spinal Ligament sponsored by the Japanese Ministry of Health and Welfare in 1993. Patients were then divided into three groups according to the maximum compression level, identified through CT and MRI scans: upper thoracic (T1–T4), middle thoracic (T5–T8), and lower thoracic (T9–T12). This categorization, based on previous research indicating better recovery in upper T-OPLL [3,20], allowed for a detailed exploration of the impact of different ossification morphologies and compression levels on the JOA score recovery rate.

### 2.4. Statistical Analysis

We used Spearman’s rank correlation coefficient to evaluate the correlation between JOA score recovery rates and kyphosis progression. To assess the correlation between the duration from symptom onset to surgery and JOA score recovery rates, we employed Pearson’s correlation test. The Tukey-Kramer test was used to identify differences in JOA score recovery rates according to ossification focus morphology and maximum compression level. Data analysis was conducted using JMP version 16 (SAS Institute Inc., Cary, NC, USA), and a *p*-value of less than 0.05 was considered statistically significant.

### 2.5. Ethical Considerations

This retrospective study was conducted in accordance with ethical standards. Given its retrospective nature, the need for individual patient consent for the use of anonymized data was waived in line with our institution’s guidelines. However, all patients provided written informed consent for the surgical procedures as part of standard clinical practice. The study received ethical approval from our university’s ethics committee on 14 March 2022, with the Approval Reference Number: M10251.

### 2.6. Manuscript Preparation and Proofreading

In the process of preparing this manuscript, we utilized an artificial intelligence (AI) tool, ChatGPT, developed by OpenAI, specifically for English language proofreading. The AI was used to assist in refining the clarity and coherence of the manuscript, correcting grammar and spelling, and enhancing the overall readability.

## 3. Results

### 3.1. Patient Demographics

This study encompassed a carefully selected cohort of 20 patients who met the eligibility criteria. A summary of the patient demographics, highlighting key characteristics, is provided in the Table 1. The average age of the patients at the time of surgery was 51 years, reflecting a population that typically presents with T-OPLL. There was a slight predominance of male patients, with 12 males constituting 60% of the sample, compared to 8 females. This gender distribution aligns with previous research suggesting a higher prevalence of T-OPLL among males. The median duration from symptom onset to surgery was found to be 7.5 months.

### 3.2. Clinical Outcomes

The preoperative average JOA score was 3.6, setting the baseline for understanding the patients’ condition before intervention. At 1 year post-surgery, the average JOA score improved to 7.4, with a JOA recovery rate of 52%. At 3 years post-surgery, the average JOA score was 7.3 with a JOA recovery rate of 53%. At 5 years post-surgery, the average JOA score was 7.3 with a JOA recovery rate of 51%. At 10 years post-surgery, the average JOA score was 7.4 with a JOA recovery rate of 51%. There was no observed correlation between the duration from symptom onset to the time of surgery and the JOA recovery rate.

Out of 20 patients, 8 patients (40%) required additional surgery, resulting in a total of 10 surgeries performed. Among these additional surgeries, 3 were performed on cases of cervical OPLL with myelopathy, and 4 were performed on cases of lumbar OPLL with cauda equina symptoms or radicular symptoms. In 1 case, the surgery addressed adjacent intervertebral disorders at the upper instrumented vertebra, and in another case, it addressed disorders at the lower instrumented vertebra. Of these 8 patients, 3 cases had a second or subsequent operation after more than 10 years. In one instance, the patient’s postoperative improvement was considered insufficient, as indicated by unchanging JOA scores, 22 months after the initial PDF. This led to additional anterior decompression surgery from the posterior approach.

### 3.3. Radiographic Evaluation

The kyphotic angle in the T4–12 region was systematically measured at various stages, displaying a progressive increase over the course of the study. Preoperatively, the angle was 26 degrees, and it increased to 29 degrees immediately post-operation. Further increases were observed at 32 degrees at 3 years post-operation, 33 degrees at 5 years post-operation, and 37 degrees at 10 years post-operation. The kyphotic angle at the fusion levels, where the preoperative value was 26 degrees, remained unchanged immediately post-operation. Subsequent measurements showed an increase to 29 degrees at 3 years post-operation, 30 degrees at 5 years post-operation, and the value was 32 degrees at 10 years post-operation. Despite the variations observed in the kyphotic angles both in the T4–12 region and the fusion levels, no correlation was found between the progression of kyphosis and the JOA recovery rates or the requirement for additional surgery.

Out of the 20 cases examined in this study, the ossification morphology was classified into specific types. Eight cases (40%) were classified as beak ossification, 4 cases (20%) as continuous waveform ossification, and 8 cases (40%) as continuous cylindrical ossification. Despite the variation in these ossification types, no significant relationship was found between the morphology of the ossification foci and either the JOA recovery rates or the need for additional surgery. Furthermore, the maximum compression level within the thoracic spine was identified in each case. Four cases were located in the upper thoracic spine, 13 cases in the middle thoracic spine, and 3 cases in the lower thoracic spine. Again, no significant relationship was found between the maximum compression level and the JOA recovery rates or the need for additional surgery.

### 3.4. Representative Case

The patient was a 22-year-old man with a diagnosis of T-OPLL at T6–9 and progressive myelopathy (as shown in Figure 2a). He underwent a T3–12 posterior decompression and fusion with instrumentation (as shown in Figure 2b). After 3 years following the initial surgery, the patient developed weakness in his left lower limb. CT and MRI scans revealed stenosis of the right and left intervertebral foramen due to L4/5 OPLL and ossification of the ligamentum flavum (OLF) as demonstrated in Figure 3a. The patient underwent an L4/5 facetectomy and resection of the OPLL and OLF (Figure 3b). After the surgery, his gait disturbance improved. After 9 years following the initial surgery, the patient developed gait disturbance. A CT scan revealed an adjacent intervertebral lesion at the upper end of the initial surgical fixation. The patient underwent posterior decompression and fixation, and the rod was extended and fixed (as shown in Figure 4). Postoperatively, the patient’s gait disturbance improved and his JOA score was 10.5/11.

## 4. Discussion

The present study examined the long-term outcomes of PDF surgery for T-OPLL in a retrospective case series of 20 patients, providing valuable insights into this complex and often debilitating condition. Our primary finding was that the mean JOA improvement rate at 10 years post-surgery was 51%, reflecting a significant and sustained reduction of symptoms. This rate, while promising, is contrasted by the observation that 40% of the cases required additional surgery. This additional intervention was mainly necessitated by systemic ossification tendencies, which can lead to the development of ossification in other levels of the spine, such as cervical or lumbar OPLL. This process may introduce new symptoms, complicating the postoperative course. Furthermore, we observed a gradual progression of kyphosis in some patients over the 10-year follow-up period. Clinicians must be prepared to closely monitor these patients in the years following surgery, administering additional interventions if needed to address the potential for gradual deterioration of the condition over time. Our study emphasizes the importance of long-term follow-up in T-OPLL, providing insights that may guide future research and clinical practice.

The choice between anterior and posterior surgical approaches often depends on the specific level and pattern of ossification. The middle thoracic level (T4/T5–T7/T8) presents unique anatomical challenges, making the posterior approach more favorable than the anterior approach [20]. The anterior approach, while direct, is fraught with technical difficulties. Notably, it poses a higher risk of complications such as spinal cord injuries, dural tears, and other associated injuries, especially when considering the direct removal of ossification foci. This risk underscores the advantages of the posterior approach, typified by PDF surgery, which is relatively straightforward and offers various decompression techniques without the need for direct intervention on the OPLL [14,20]. Our research, along with other studies [3,13], has shown that PDF surgery can achieve significant neurological recovery, even when anterior impingement by residual OPLL remains. Given the complex nature of T-OPLL, a patient-specific treatment strategy is essential. Various surgical techniques, whether anterior, posterior, or a combination of both, have been explored [1,10,21,22,23,24,25,26,27,28,29]. However, determining the most effective approach for T-OPLL patients, balancing both efficacy and associated risks remains a subject of ongoing research.

Our findings suggest that the outcomes of PDF for T-OPLL are largely maintained over a 10 year period, aligning with previous research that has observed similar stability during long-term follow-up [17]. In the referenced study, kyphosis correction was a component of T-OPLL surgery, leading to JOA improvement rates of 55.1%, 54.8%, 53.0%, and 51.9% at 1, 2, 5, and 10 years postoperatively. However, a gradual decline in long-term outcomes can occur, particularly in cases where systemic ossification tendencies lead to the development of ossification in other levels of the spine [2,7,30,31,32,33]. This process can appear as new symptoms and may complicate the postoperative course. Our study’s observations reflect these insights, with 35% of cases requiring surgeries for cervical or lumbar T-OPLL, a figure that is comparable to previous research reporting an additional surgery rate of approximately 17.6% after PDF for T-OPLL [17]. While the posterior approach, particularly PDF surgery, has been underscored as a preferred method due to its advantages and reduced risks, it is essential to understand its potential limitations. In some rare instances, despite the generally favorable outcomes of PDF surgery, the desired symptom relief may not be achieved. This is particularly observed when there’s no notable postoperative improvement, as indicated by stable JOA scores. Such scenarios might necessitate additional interventions such as anterior decompression. Although PDF surgery for T-OPLL can be highly effective, there remains a potential drawback. In some cases, the symptom relief achievable through anterior decompression might not be attained due to the persistent spinal compression from the ossified lesions [3]. In these nuanced situations, clinicians must consider the potential benefit of incorporating anterior direct removal or anterior decompression of the spinal cord, especially if symptoms persist post-PDF surgery.

The results of kyphosis angle progression observed In our study following T-OPLL surgery were consistent with findings from previous research [1,16]. Our research, which included thoracic instrumented fusion, adds to this body of knowledge, demonstrating a gradual progression of kyphosis over an extended 10-year follow-up period. Specifically, we observed an average increase of 8 degrees in the T4–12 region and an average of 6 degrees in the fused level from the immediate postoperative period to 10 years after surgery. Interestingly, our findings align with previous research in revealing that kyphosis progression, despite its measurable increase, did not negatively impact key clinical outcomes such as JOA scores or the need for additional surgeries [1,16]. However, our study also emphasizes the importance of careful interpretation. The absence of immediate clinical impact from kyphosis progression should not be misinterpreted as a lack of clinical significance. Close monitoring of kyphosis and proactive management must still be integral to the long-term follow-up of T-OPLL surgery patients. Incorporating regular assessments and potential interventions into the standard care pathway can enable early detection and management of new lesions in different levels of the spine, further enhancing the long-term success of T-OPLL treatment. This comprehensive approach ensures the overall well-being and quality of life of patients in the long term, reflecting the complex and multifaceted nature of T-OPLL management.

Our study presents several limitations that are essential for readers to recognize, each providing directions for future research. A primary limitation stems from our retrospective design. This design inherently introduces certain complexities, including the variability in pre-surgical JOA scores among patients. Although some studies have posited that shorter durations from symptom onset to surgery can result in better JOA improvements, our findings did not explicitly corroborate this [34,35]. Notably, in real-world clinical settings, a subset of patients often defer seeking medical attention until a significant duration has passed from symptom onset, usually when their myelopathy has considerably worsened. This trend towards delayed medical consultation might have contributed to the diversity we observed in pre-surgical JOA scores. The second limitation was our small sample size, an issue partly attributed to the rarity of T-OPLL. Including only 20 patients may present challenges in extending our findings to the broader population of patients with T-OPLL, supporting the need for future studies with larger and more diverse patient populations. The third limitation was the relatively low tracking rate, with some patients lost to follow-up, a situation that can be understood considering the extended period of 10 years or more involved in the study. This natural challenge in long-term studies could introduce bias into the findings, emphasizing the need for strategies to maintain consistent follow-up in future research. Additionally, a lack of comparative analysis with other treatment modalities or demographic groups might have provided further insights. Further studies integrating these comparative aspects can contribute to a more nuanced understanding of PDF surgery for T-OPLL. Despite these limitations, our study is one of the few to report long-term results over 10 years, laying a foundation for further exploration and offering essential insights to guide subsequent research.

## 5. Conclusions

In summary, the present study evaluated the long-term outcomes of PDF surgery for T-OPLL in a retrospective case series of 20 patients. The results of the study showed that the mean JOA improvement rate at 10 years post-surgery was 51%. Although 40% of the cases required additional surgeries mainly for cervical or lumbar OPLL or adjacent segment disorders, the overall 10-year outcomes of PDF were generally favorable. Additionally, we observed a gradual progression of kyphosis in some patients over the 10-year follow-up period, however, our findings showed that kyphosis angle progression did not negatively impact clinical outcomes. Our findings suggest that PDF surgery is a suitable treatment option for patients with T-OPLL and highlights the importance of long-term follow-up for these patients.

## Figures and Tables

**Figure 1 jcm-12-05701-f001:**
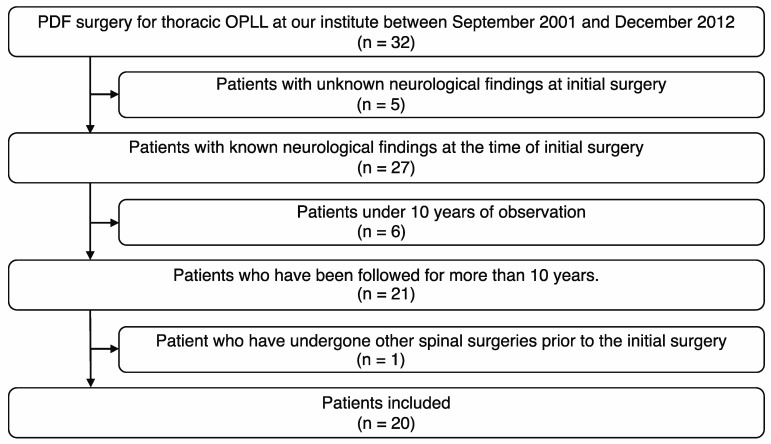
The flowchart detailing the inclusion and exclusion criteria for patient selection.

**Figure 2 jcm-12-05701-f002:**
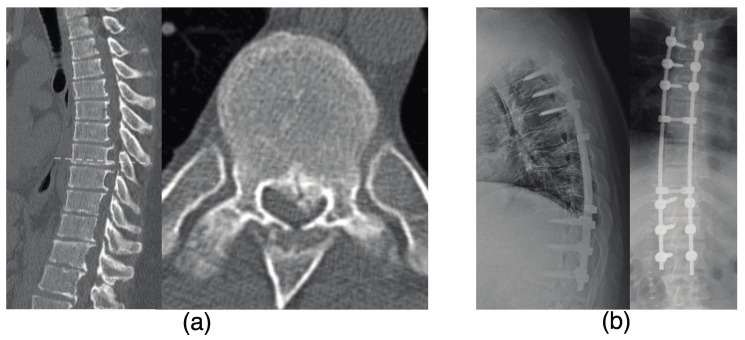
The 22-year-old male patient presented with progressive myelopathy due to thoracic ossification of the posterior longitudinal ligament at T6–9 and progressive myelopathy (**a**) who underwent T3–12 posterior decompression with instrumented fusion (**b**). The preoperative JOA score was 5, and at 1 year post-surgery it was 9.5, with a recovery rate of 75%.

**Figure 3 jcm-12-05701-f003:**
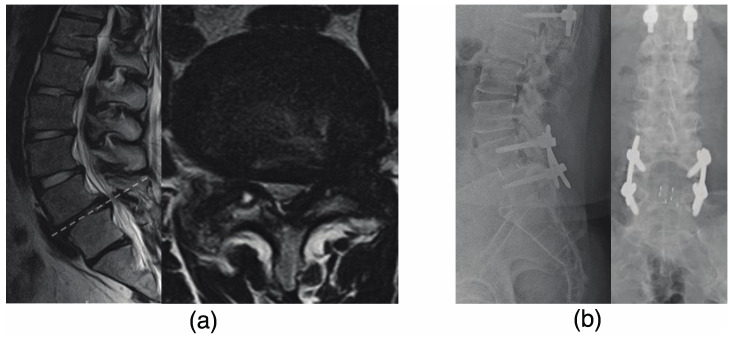
Three years after the surgery, the patient developed weakness in his left lower limb. (**a**) An MRI revealed stenosis of both the right and left intervertebral foramen caused by L4/5 OPLL and OLF. (**b**) The patient underwent an L4/5 facetectomy and resection of the OPLL and OLF. MRI, magnetic resonance imaging; OPLL, ossification of the posterior longitudinal ligament; OLF, ossification of the ligamentum flavum.

**Figure 4 jcm-12-05701-f004:**
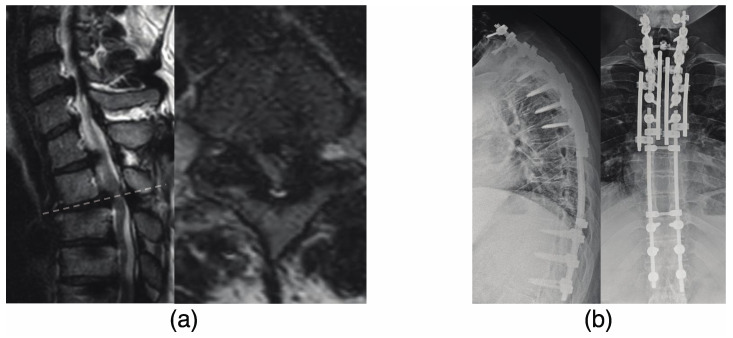
(**a**) The patient developed gait disturbance 9 years after the initial surgery. An MRI scan revealed an adjacent intervertebral lesion at the upper end of the initial surgical fixation. (**b**) The patient underwent a posterior decompression with instrumented fusion, and the rod was connected to the previously implanted rod.

**Table 1 jcm-12-05701-t001:** Patient Demographics.

	Age	Sex	BMI	Comorbidity (DM, HT)	Maximum Compression Level	Location of OPLL	Spinal Canal Occupancy Ratio (%)	Type of OPLL	OLF	Duration of Symptoms (Month)	Preoperation JOA Score
1	74	Female	20		middle	T6–8	58	continuous cylindrical		72	6.5
2	52	Female	26	DM, HT	middle	T5–7	50	continuous cylindrical		2	4
3	41	Female	35	DM	upper	T3–7	42	continuous waveform		4	6
4	55	Male	27	DM	middle	T6–9	36	continuous waveform	positive	4	3
5	65	Female	27	DM, HT	middle	T5–8	43	continuous cylindrical	positive	8	5.5
6	60	Male	27	DM, HT	middle	T4–7	79	beak	positive	24	6
7	64	Male	27	HT	upper	T1–3	50	continuous cylindrical	positive	5	3
8	59	Male	32	DM, HT	middle	T6–9	62	beak	positive	9	2.5
9	42	Male	32	DM	middle	T9–11	46	beak		7	2.5
10	53	Female	35		middle	T6–8	61	continuous cylindrical	positive	3	1.5
11	68	Male	22	DM	middle	T4–5	72	beak	positive	13	1.5
12	37	Male	34	DM, HT	middle	T4–7	40	beak	positive	7	1
13	34	Male	45	DM	lower	T8–11	45	continuous waveform	positive	11	4.5
14	49	Male	30	HT	upper	T2–5	48	beak	positive	20	1
15	35	Male	37		upper	C3–T5	51	continuous cylindrical		4	3.5
16	72	Female	24	DM, HT	lower	T8–L3	60	continuous cylindrical		21	3.5
17	22	Male	43	DM	middle	T6–9	60	continuous cylindrical		7	5
18	36	Male	27	DM	lower	T8–L2	53	beak		9	4
19	48	Female	35	DM	middle	T5,6	57	beak	positive	7	2.5
20	59	Female	22		middle	T5–10	54	continuous cylindrical		17	4

BMI, body mass index; DM, diabetes mellitus; HT, hypertension; OPLL, ossification of the posterior longitudinal ligament; OLF, ossification of the ligamentum flavum; JOA, Japanese Orthopedic Association. Patients were categorized based on the maximum compression level of their thoracic spine into three groups: upper thoracic (T1–T4), middle thoracic (T5–T8), and lower thoracic (T9–T12).

## Data Availability

The data generated during this study are not publicly available due to privacy or ethical restrictions.

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
