# Peer review of "Posterior Decompression and Fixation for Thoracic Spine Ossification: A 10-Year Follow-Up Study"

_jcm, 2023, doi:10.3390/jcm12175701_

Round 1

Reviewer 1 Report

Thank you for the opportunity to review the paper with the title 'Posterior Decompression and Fixation for Thoracic Spin oOssification: A 10-Year Follow-Up Study'. 

It is a useful paper as Ossification of Posterior Ligament is a rare condition, could lead to potentially devastating events with neurological deficits and immediate treatment strategy should be designed. Posterior decompression and fusion is an effective approach, not technical demanding with less complications compared to anterior or combined approach. The findings of the study are consistent with the current literature, there is a good use of the English language and the paper is well stuctured.

However there some issues that need to be addressed before publication of the paper.

1. Due to the small number of patients of the study the results could not be generalised. The statistical analysis is weak. The authors should provide additional tables and statistical analysis regarding postoperative kyphotic angle in relation with timing of symptoms, surgery and recovery rate. 

2. Did the patients had been reviewed by rheumatologists?

3. The authors should compare the complication rate of posterior approach in the current study with previous studies as well as with anterior or combined approach.

4. The neurological status of patients pre and post-operatively should be included.

5. In Abstract line 26: Cervical or lumbar OPLL is not diagnosis, it is a radiological finding. Myelopathy or nerve compression for example are conditions that could lead to surgery.

6. In Introduction section, lines 35-36: T-OPLL is a radiological finding that could cause spinal cord or spinal nerve compression. These findings lead to the symptoms mentioned, please modify.

7. In Introduction section, lines 41-44: Posterior decompression and fusion is well known for its effectiveness for thoracic decompression and multiple studies have been performed regarding this matter. I assume that the authors would like to highlight the superiority of the above method (posterior approach) compared to the anterior or combined approach.

8. In Introduction section, lines 45-46: This condition is usually referred as DISH. Is this condition the one that authors describe or any other rheumatologic condition?

9. In Material and Methods section, line 77: Is Hirabayashi's method a modified JOA score?

10.  In Material and Methods section, line 83:   Is there any data regarding the initiation of symptoms and the time until surgery is performed? This information is crucial for the recovery rate.

11. In the Results section, lines 129-130: Please define the condition that led to the surgeries. Was there myelopathy for the cervical spine or central canal stenosis with neurogenic claudication or neurological deficits for the lumbar spine?   12. In Discussion section, line 212: Please define optimal neurological recovery.   13.  In Discussion section, lines 218-220: How did the authors characterized that improvement is insufficient?

Reviewer 2 Report

Dear authors i am pleased to review your article. It is written well. some monor changes required please see the attach file for comments.

Thanks

Round 2

Reviewer 1 Report

Dear authors, I find your responses quite satisfactory. You addressed all the issues raised in the previous round of the review process. I suggest your paper can be published in its current form.